# Clinical Supervision across Australia, Türkiye, Syria, and Bangladesh: From WEIRD to WONDERFUL

**Salah Addin Lekkeh [1], Md. Omar Faruk [2,*], Sabiha Jahan [2], Ammar Beetar [1], Gülşah Kurt [3] , Ruth Wells [3] and Scarlett Wong [3]**

[1]   Hope Revival Organization, 27090 Gaziantep, Turkey
[2]   Department of Clinical Psychology, University of Dhaka, Dhaka 1000, Bangladesh
[3]   School of Psychiatry and Mental Health, University of New South Wales, Sydney 2052, Australia
*    Correspondence: orhaanfaruk07@gmail.com

**Abstract: Background:** Clinical supervision in providing mental health and psychosocial support services (MHPSSs) is an ethical imperative and a key to ensuring quality of care in terms of service users' skills enhancement, well-being, and satisfaction. However, humanitarian contexts in low-resource countries usually lack sufficient infrastructures to ensure staff have access to supervision. Against this backdrop, a pilot supervision program was introduced in Bangladesh and Syria to help MHPSS staff provide quality care. However, supervision provided by experts unfamiliar with these contexts decontextualizes the supervision process and hinders cultural relevance. The aim of this paper is to present a decolonial model of supervision called "WONDERFUL Supervision". **Methods:** We provided fortnightly online supervision to a total of 32 MHPSS practitioners (seven in Bangladesh and twenty-five in Syria) working in humanitarian contexts in Bangladesh and Syria as well as their surrounding countries (such as Türkiye) between 2019 and 2021.The issues talked about were the skills needed for the practitioners to provide optimal levels of service, manage staff burnout, and present cases. Focus group discussions and reflective discussions included 19 participants, involving both practitioners and supervisors across sites. **Results:** Despite some notable effects, the supervision was obstructed due to being decontextualized, such as the supervisors not having adequate knowledge about the contexts and culture of beneficiaries, a perceived feeling of power imbalance, practitioners having limited access to resources (e.g., internet connection and technical support), and different time zones. This defect paves the way for a new mode of supervision, WONDERFUL, which takes into account contextual factors and other sociocultural aspects. **Conclusions:** WONDERFUL supervision has the potential to indigenize the concept of clinical supervision and thereby more sustainably and effectively ensure quality mental health care in resource-limited countries, especially in humanitarian contexts.

**Keywords:** clinical supervision; WEIRD; WONDERFUL; Australia; Turkey; Syria; Bangladesh

## 1. Introduction

*Clinical Supervision as a Form of Collective Healing in Humanitarian Settings*

People affected by conflicts are at greater risk of developing mental health problems, such as depression, anxiety, and post-traumatic stress disorder (PTSD) (Charlson et al. 2019). The increasing demand for mental health and psychosocial support (MHPSS) in humanitarian settings has necessitated scalable solutions, such as training non-specialist workers to deliver brief psychosocial interventions (Sijbrandij et al. 2017). These workers can be at a higher risk of mental health problems, suffering from the adverse impacts of anxiety, depression, PTSD, secondary stress, burnout, and substance abuse (Curling and Simmons 2010). The integration of regular clinical supervision can protect the well-being of MHPSS workers and increase service user satisfaction (Perera et al. 2021), especially in humanitarian settings where the practitioners themselves are exposed to distressing environments

(Böhm et al. 2022). However, supervision has received scant attention in humanitarian contexts in low- and middle-income countries (LMIC).

Local MHPSS staff members working in humanitarian settings are vital to collective healing because they are part of the local community, they understand the local culture and context, and they are likely to remain after international staff leave. Supervision provides an opportunity to support these members of staff to alleviate the psychosocial suffering of members of their community. Clinical supervision is integral to clinical skills development and competency (Barrett et al. 2020; Böhm et al. 2022; Falender and Shafranske 2017). However, access to experienced psychologists for supervision, support, and mentoring can be difficult in conflict settings. Video conferencing is a practical, accessible, and cost-effective solution to provide support to psychosocial workers working in often highly stressful, yet minimally supported environments (Kemp et al. 2019).

A group of Australian psychologists connected online with groups of Syrian and Bangladeshi mental health workers to provide them with clinical group supervision from 2019 to 2021. The supervision included developing the clinical skills needed to deal with clients from different cultural contexts and to manage the mental health issues of those who are providing clinical services in both settings. In each setting, experienced Australian supervisors offered the supervision. The duration for each session was one and a half hours fortnightly. Practitioners who consented to participate in the pilot program were selected. The findings from this program were used to develop a subsequent scaled-up program—the Caring for Carers (C4C) program. Participatory research and equal representation are crucial ethos of the C4C program. The program aimed to test the effectiveness and acceptability of an online clinical supervision program in the humanitarian contexts of Bangladesh, Northwestern Syria, and Türkiye to improve the well-being of practitioners, increase their clinical skills efficacy, and improve service quality and satisfaction. The C4C program consisted of two terms of eight-month-long supervision sessions provided via an online platform on a fortnightly basis to a group of MHPSS practitioners. Each session lasted 90 min. One Australian and one local supervisor (Bangladeshi or Syrian) co-facilitated the supervision group of four to six MHPSS practitioners. The program focused on reflective practice, case discussion, and skills learning and improvement while providing MHPSSs to the displaced communities in the respective context. When the C4C clinical supervisor lead from Australia (SW) was asked to present at the 11th ISHHR Conference for Health and Human Rights about the supervision intervention, we decided to develop a collaborative format so that our findings would equally represent clinicians from the three communities involved in the supervision: Syria, Bangladesh, and Australia. We all bring professional as well as personal lived experience from our respective contexts. The core group working on this paper includes a Syrian psychologist living in Türkiye (SL), two Bangladeshi clinical psychologists in Dhaka, Bangladesh (SJ and OF, SJ is also a PhD student), and an Australian clinical psychologist and supervisor (SW, also a PhD student). This group undertook weekly reflective meetings to think critically about how coloniality impacted on the initial program. The remaining Syrian (AB), Turkish (GK) and Australian (RW) authors supported the editing of this manuscript. This paper describes our findings from delivering a cross-cultural, online group supervision with mental health workers working in humanitarian settings. We also describe the process we undertook to work towards transcultural understanding and decolonial practice. We begin by providing some contextual information about the settings we work in and explain why a decolonial approach is so important to mental health practice in humanitarian settings.

## 2. The Settings We Work In

### 2.1. Humanitarian Context in Cox's Bazar

More than 1 million Rohingya people fled Rakhine State in Myanmar and sought refuge in Bangladesh's Cox's Bazar amid the brutal military crackdown orchestrated by the Myanmar Army in 2017. The United Nations has deemed it as a textbook example of ethnic cleansing (United Nations 2017). The Rohingya people have endured decades of

discrimination, widespread human rights abuses, persecution, and violence. The Rohingya people in Bangladesh are forced to live in over-populated, squalid camp situations and lack access to safe drinking water, sanitation, and other healthcare services. This can contribute to mental health problems, especially in children and the elderly. Humanitarian contexts in Bangladesh largely suffer from a lack of consideration in ensuring service providers' well-being, insufficient clinical supervision, and a lack of skilled service providers (Islam and Mozumder 2021). Moreover, the majority of the organizations providing MHPSSs in humanitarian contexts do not have provisions for clinical supervision partly due to the lack of awareness of its potential benefits. Moreover, the lack of mental health professionals with experience of clinical supervision has also contributed to the gap.

Cox's Bazar has a reputation of being one of the most famous tourist attractions in Bangladesh. Yet, Cox's Bazar is considered one of the poorest cities in Bangladesh and is highly susceptible to climate disasters. Approximately 33 per cent of the total population lives below the poverty line (Bangladesh Bureau of Statistics 2017). The massive influx of the Rohingya people has created a humanitarian crisis in this region, with approximately 1.2 million people in host communities, especially those in Ukhiya and Teknaf, adversely affected. The increased attention paid to ensuring the basic rights of the Rohingya people has left host communities with inadequate access to health, education, and employment.

## 2.2. Humanitarian Context in Northwest Syria and Türkiye

The conflict in Syria has led to the displacement of more than 13 million people since 2011. A total of 6.6 million people sought refuge in neighboring and Western countries while 6.5 million were forcibly displaced within their country. Many of those internally displaced live in the camps in Northwest Syria with limited access to basic services. A total of 3.7 million Syrians live under temporary protection status in Türkiye, which became the top host county for refugees in the world (UNHCR 2022). Syrians have experienced many conflict-related traumatic incidents and post-displacement stressors in their new resettlement contexts. Given the multitude of stressors, they are at a high risk of developing mental health disorders (Kurt et al. 2022). Yet, access to mental health services in both Northwest Syria and Türkiye is very limited because of an insufficient number of experienced MHPSS staff. Our consortium has also recently highlighted a need for ongoing clinical supervision to improve the MHPSS provision in Türkiye and Syria (Wells et al. 2020). However, similar to the situation in Bangladesh, supervision is often overlooked because of restrictions for funding and the lack of awareness about its potential benefits.

## 2.3. WEIRD Psychology

Emerging from Western countries, such as the UK, France, Germany, Netherlands, Australia, Canada, and the US, the field of psychology has been built on Western values, practices, and epistemic (meaning-making) systems (Newnes 2021). Psychology, as part of the social sciences, has developed from research on WEIRD populations, meaning those who are Western, educated, industrialised, rich, and democratic (Roberts et al. 2020). The majority of psychological research is dominated by the inclusion of white subjects, without attending to the impact of racial and cultural diversity (Roberts et al. 2020; Henrich et al. 2010a). It is argued that basic cognitive and motivational processes vary across cultures, with evidence suggesting that people of different ethnic backgrounds tend to explain and predict behaviour differently (Henrich et al. 2010a). Therefore, findings generated from a respective setting may not represent people from various cultures, putting a question mark on how universal generalisations may be (Henrich et al. 2010b). Against this backdrop, growing attention has been paid on deconstructing and localizing research by embracing indigenous perspectives and ideas. By privileging Western epistemic systems, psychology may contribute to epistemic injustice if it is applied to people from diverse backgrounds without a critical examination if how this may marginalise other meaning-making systems. Epistemic injustice is when certain kinds of knowledge (e.g., Western concepts of individualism) are given more value than other kinds (e.g., Syrian concepts of family connectedness) in ways

that can marginalise certain groups of people. The implications of WEIRDNESS can be observed in the supervision process too. Supervision provided by Western-based supervisors is likely to overlook contextual factors at play, thereby undermining the cultural relevance and utility of supervision for MHPSS staff.

*2.4. Aims*

In this paper, we aim to bring together Bangladeshi, Syrian, and Australian perspectives to develop a decolonial model of transcultural supervision. In particular, we take a critical approach to WEIRD psychology to interrogate its impact on supervision in our pilot programs. We then outline an approach to overcome some of these challenges.

**3. Methods**

We followed a participatory research approach that focuses on the process of sequential or successive action and reflection involving local people. This approach draws on local knowledge and viewpoints that are essential to forming the basis of research and planning (Cornwall and Jewkes 1995). In the present study, we engaged in two forms of knowledge production: focus group discussions and reflective discussions. Firstly, we conducted a series of stakeholder workshops with practitioners involved in the pilot as part of the C4C program. Second, we engaged in a series of regular structured meetings which involved reflective conversations aimed at giving equal weight to all voices in the conversation. We used these conversations to synthesise information from the stakeholder workshops with our professional, cultural, and lived experience to generate a new model for supervision in humanitarian settings.

*3.1. Focus Group Discussions*

3.1.1. Participants

Nine MHPSS practitioners (six male and three female) in Northwest Syria and Türkiye and six (four female and two male) in Bangladesh attended the focus group discussions. The average age of practitioners was 44 in Northwest Syria/Türkiye and 32 in Bangladesh. All took part in the pilot supervision program between 2019 and 2021.

3.1.2. Materials (i.e., Interview Guides)

Focus group discussion guide was prepared to uncover the experiences of pilot program participants. Discussion questions focused on the most and least useful aspects of the pilot program, targeted clinical competencies, areas of improvement, barriers, and enablers to implement supervision program as a regular practice in MHPSS field.

MHPSS practitioners were asked to attend the focus group discussions. They were presented with an informed consent form explaining the terms and conditions of attending the discussion. Due to COVID-19 pandemic, they were offered to attend the discussion either online or in person. The workshop was conducted in the native language of the participants (Arabic for Syrian supervisors and Bangla for Bangladeshi supervisors). Each workshop lasted around two hours. Once the discussion session was completed, they were asked to answer questions related to sociodemographic characteristics. Practitioners were given contact details of the researchers if they wanted to be informed about the discussion results.

Focus group discussions (two in Northwest Syria/Türkiye and one in Bangladesh) were audio recorded and transcribed into Arabic and Bangla. Approximately six hours were recorded. The transcriptions were translated into English to work collaboratively in analysing discussion data. The final transcripts were 10 pages long for Syrian practitioners and 12 pages long for Bangladeshi supervisors.

## 4. Analysis

A thematic analysis (Braun and Clarke 2006) with elements from grounded theory (Charmaz 2014) was conducted to identify the common themes in the focus group discussions.

### 4.1. Reflective Discussions

Participants

Four members of the C4C team meet weekly for 10 weeks for 90 min for each meeting as part of the reflective discussion. These meetings included the following members:

The first is a Syrian psychologist who has been working in the humanitarian mental health field for a decade in Syria, Lebanon, and Türkiye. He has lived in South Türkiye since 2019 and works in Türkiye/Northwest Syria. He (as a refugee) works as a psychologist, specialist, trainer, and supervisor providing MHPSSs for Syrian refugees and internally displaced people. He was a part of the pilot supervision program between 2020 and 2021. In 2022, he started working as a co-supervisor for the C4C project and as a research investigator. He is interested in research and practice related to supervision and scalable psychosocial interventions.

The second is a Bangladeshi clinical psychologist who has more than 15 years of clinical experience in both practice and supervision. She has recently been trained in the reflective supervision model through the integrated model of supervision, which has enriched her contributions to this program. Her personal role in the project was to reflect the ongoing conversations and name any issues that emerged in the collaborative discussions, which she achieved through systemic and narrative training.

The third is a Dhaka-based clinical psychologist who currently oversees C4C field research activities in Cox's Bazar, Bangladesh. He is interested in the deconstruction of the colonial approaches that have shaped mental health practices over the years in low-resource countries, especially in humanitarian contexts. He is keen on the inclusion of underrepresented groups irrespective of race, gender, and nationality in his research and the transference of indigenous knowledge into practice.

The fourth is an Australian clinical psychologist who has many years of experience as a clinical supervisor in Australia, as well as in humanitarian settings. She is the clinical lead of the C4C project and a PhD candidate at UNSW Sydney. She is from a Chinese-Vietnamese refugee background and, as such, is sensitive to the role of power dynamics and epistemic injustice based on ethnicity, gender, culture, and nationality.

The reflective discussions involved a collaborative analysis, drawing on the clinical experience of the group members, which resulted in the generation of the model.

## 5. Results

### 5.1. Reflective Practice

5.1.1. Clinical Supervision in Bangladesh

Applied branches of psychology in Bangladesh began with the establishment of the Department of Clinical Psychology, University of Dhaka in 1997. Since its inception, the department has been providing both individual and group supervision to trainees by qualified supervisors. Besides clinical psychology, its other applied branches also provide clinical supervision. The Department of Educational and Counselling Psychology (established in 2011) provides individual supervision to trainees. Master's programs, such as industrial psychology and school psychology, under the Department of Psychology also provide supervision. However, the mode of supervision depends on the theoretical orientation of the applied branches. For example, specialized psychological services, such as systemic family therapy, psychodrama, trauma-informed therapy, and neuro-linguistic programming, include supervision from respective experts. Besides academic supervision, applied branches also provide supervision to paraprofessionals who receive basic counselling skills training from these departments. In addition to that, several non-governmental organizations in Bangladesh offer training courses that often include the provision of clinical supervision. Clinical supervision in humanitarian contexts, on the

other hand, is scarce and sometimes is not prioritized in the MHPSS provision. However, some non-governmental organizations provide supervision related to line management, which is often confused with clinical supervision.

### 5.1.2. Clinical Supervision in Syrian Arab Republic

In 2010, before the Syrian crisis, there were 120 Syrian psychiatrists. A total of 80% of them were in the capital of Syria, Damascus (Hedar 2017). Psychiatrists were allowed to provide psychiatric services in Syria. Until now, there has been no legal licensing system for psychotherapists, psychologists, psychiatric nurses, or social workers in Syria (Eloul and Quosh 2013). Up to ten psychologists received training abroad in clinical psychology. Syrian universities offer 5 years of academic courses in psychological counselling and psychology (Hedar 2017), mostly without any practical training or supervision. However, reviewing the MHPSS literature in Syria before and after 2011, we came across a paper that included the concept of supervision; it was published by international experts who worked for UN agencies in Syria to support Iraqi refugees after 2003 (Quosh 2013). That being said, after 2011, another paper published by the Syrian Psychiatric Association did not include supervision or refer to it at all (Hedar 2017). However, some groups of psychologists have organized informal support groups which can be counted as a form of supervision.

Due to the accelerated conflict in Syria in 2011, the MHPSS field underwent a series of transformations. In response to the shortage of specialized mental health professionals in this setting, psychosocial practitioners from various professional backgrounds were trained on delivering brief, scalable psychosocial interventions, such as Problem Management Plus (PM+) (Nemiro et al. 2021). As part of these training sessions, they received regular supervision to discuss implementation-related challenges. In addition, a group of Syrian MHPSS practitioners received focused training on the systematic supervision approach called the Holloway Model (Holloway 2016), which was provided by a health-focused capacity-building group in Türkiye. These developments in the field attracted scholarly attention and highlighted the importance of supervision as a part of MHPSS practice. As supervision is relatively new in Northwest Syria and Türkiye, there is limited evidence to determine the effectiveness of supervision in regular practice.

### 5.1.3. Australian Supervision Context

Australian psychologists have had the benefit of a stable and high-income contextual environment with a regulated, well-resourced, and accessible education system that supports psychology as a profession. In order to become a psychologist in Australia, there is a standardized and regulated pathway that is monitored by a governmental regulator, the Australian Health Practitioner Regulation Agency (AHPRA). This ensures that not only beneficiaries of psychological services are protected, but also Australian psychologists themselves are emotionally and professionally protected through regular mandated clinical supervision. The title of psychologist is protected by law and can only be used by professionals who have had recognized tertiary training. As such, Australian psychologists have a relatively homogenous range of training and educational backgrounds relative to their Syrian mental health colleagues. Additionally, Australian psychologists have a relatively homogenous and standardized pathway in relation to the requirement of fortnightly clinical supervision as a part of the process of accreditation as a psychologist. After attaining professional registration, Australian psychologists are required to attain 40 h of continuing professional development, of which 10 h must be supervision, a year. As such, clinical supervision is embedded in the regulation, standards, and culture of the Australian psychology profession. Australian psychologists are accustomed to receiving and delivering reflective supervision and based on developmental models of supervision due to their training and educational culture (Gonsalvez et al. 2017; Robinson et al. 2019; Senediak and Bowden 2007). The culture and practice of reflective supervision is so embedded within the Australian psychology profession and ethos that when we designed

the pilot program, we ethnocentrically regarded this as not *a* way to provide supervision, but the way to provide supervision. However, as is outlined by our colleagues in the Bangladeshi and Syrian contexts, the Australian way of providing supervision was met with a mixed response.

*5.2. WEIRD Supervision in Bangladesh and Syria*
WEIRD Supervision in Bangladesh and Its Impact

As the clinical supervision was provided by supervisors residing in developed countries, such as Australia, there were a number of challenges that characterized the concept of WEIRD psychology. The pilot program ran a number of supervision sessions conducted by Australian supervisors having insufficient or little knowledge about the local context (e.g., indigenous ways of symptom manifestations, ignorance/indifference to cultural beliefs/customs/practices/ways of healing, and faith healers, etc.). Unfamiliar with the local contexts, it is reasonable to believe that supervisors involved in the sessions might miss some of the pertinent issues deemed important to consider by practitioners working in the context.

> *Sometimes I thought they (referring Australian supervisors) were missing points which I wanted to highlight more. This is because what is important for me and this context where I work might not seem important for them.* (Psychologist)

In addition, this can pose a barrier to effective communication between supervisors and supervisees with an additional hindrance to the exchange of knowledge. Language barriers seemed to have a far-reaching impact on the supervision process. While some of the local practitioners had a good or moderate level of English, the majority of practitioners experienced speaking English as a significant barrier for effective communication. The Australian supervisors were believed to have possessed more power than local practitioners in terms of their level of knowledge, access to resources, and being native in English language. This fostered a sense of perceived powerlessness among the local practitioners.

> *They have easy access to resources, and they do not have to wait for it. This helps them gather knowledge in a very short span of time. On the other hand, in Bangladesh, we need to ask the higher authority and wait for an indefinite period of time to arrange training or getting us access to resources.* (Psychologist)

Having different time zones (Australia is 5 h ahead of the local time in Bangladesh) interfered with office hours during the working period and prayer time especially on Fridays (Friday in Bangladesh is a weekend and a special prayer is offered).

The nature of the supervision was reported as an additional barrier to the supervision. While some local practitioners preferred to have formative supervision, others preferred restorative supervision. Formative supervision in clinical settings refers to the regular review of work and exploration of professional relationships. This can often involve discussing theories, identifying ways to integrate theories into practice, and identifying transference and countertransference. In a nutshell, formative supervision focuses on the growth of supervisees in terms of developing knowledge, skills, and abilities. It encourages supervisees to become more reflective of their practice (Proctor 2001). Restorative supervision, on the other hand, also draws on reflective practice by increasing the skills of supervisees to become more resilient in their needs and ensuring their well-being and motivation.

> *Sometimes we need to know how we can manage our issues (e.g., extreme workload, fatigue, and burnout) emerging from clinical work. Paying more attention to beneficiaries' mental health issues might also put us at risk (of developing mental health issues). There was a gap in balancing imparting skills and managing personal issues.* (Psychologist)

It was not possible for the Australian supervisors to blend either type of supervision into the process. Access to internet and devices was also reported as a major barrier. Working areas, particularly the camps where the Rohingya people reside, were not equipped with uninterrupted internet connectivity and many organizations did not provide the local practitioners with devices. Therefore, the internet connectivity was intermittent and technical difficulties interrupted the sessions, resulting in the low turnout of the local practitioners.

The supervision period was deemed to have been very short considering the gravity of issues that needed to be discussed in the supervision session. Furthermore, the sessions did not take into account special cases, gender-based violence, for example, which is very prevalent in the camps. Not all practitioners had the skills to deal with such cases.

Discussions over a single case took up a good deal of time and resulted in a perceived feeling of neglect. The supervision sessions were thought to have lacked structure (e.g., setting an agenda, future course of action, etc.). Finally, the local practitioners felt they needed additional resources to be more skilled, which the Australian supervisors did not pay any heed to.

### 5.3. WEIRD Supervision in Syria and Its Impacts

Similar to the findings in the Bangladeshi context, the pilot program revealed the importance of cultural familiarity and knowledge in the supervision process. Limited knowledge about Arabic culture and Islam was a significant barrier to pinpointing the contextual factors precipitating or maintaining mental health problems among Syrian displaced people. The lack of knowledge about different conflicts and displacement stressors in Syria and Türkiye might have misled Australian supervisors to evaluate cases in the same contextual frame. This hindered the full comprehension of the cases presented by the practitioners and thereby prevented an effective case discussion. Further, differences in communication styles were reflected in the practitioners' expectations from the supervision. Syrian practitioners are used to the directive style, in which a supervisor tells them what to do. On the other hand, Australian supervisors are used to the reflective style, in which the supervisor encourages you to come up with new perspectives and solutions. This difference led to tension between the practitioners and supervisors in the initial stage of the supervision.

In line with the pilot program findings, the results from the focus group discussions showed that the supervisors' cultural and contextual knowledge was pivotal to ensuring the utility of supervision for the Syrian practitioners. Building trust appeared to be the backbone of a healthy working relationship between the supervisors and practitioners.

> *Later, other participants were added to the group, and there was no harmony. I didn't feel comfortable in the group. I didn't attend supervision sessions due to lack of commitment to the group and safety.* (Psychological counsellor)

It was difficult for the practitioners to talk about the challenges that they encountered in their practice. Difficulty in communicating in English accelerated communication difficulties. This created power imbalances between the supervisors and practitioners.

> *Translation needs to be worked on, and in some sessions, I had to present the case and translate it, and it was exhausting.* (Psychotherapist)

Although the practitioners found learning new perspectives and methods very useful, differences in styles hampered them from seeing the utility of supervision in the beginning.

> *It was an unconventional supervision, in which there is sharing and exchange of information, and there is no authoritarian situation, and we are here to learn.* (MHPSS field officer)

After building a trusting relationship, the Syrian supervisors embraced the reflective practice and appreciated its benefits.

> *The supervision was provided in a different way than I previously attended. It was educational focusing on cases, understanding and presenting them, and intervention. And you feel empowered because it is educational supervision, and this also includes evaluating treatment plan.* (Psychotherapist and Supervisor)

In addition to the challenges related to the supervision process, organizational support was presented as an important enabler of the supervision. The lack of time and space allocated for supervision disrupted the attendance of the practitioners. Access to a stable internet connection and private space hindered practitioners' ability to attend and benefit fully from the supervision.

*5.4. Suggestions for Improvement*

5.4.1. How to Make Supervision More Familiar for Bangladesh

The integration of concepts or ideas not familiar within a targeted environment may cause them not to be accepted. However, a gradual integration of ideas may serve better in this respect and eventually they may be acknowledged. Psychologists in humanitarian contexts in Bangladesh largely lack a general understanding of supervision's pragmatic usefulness. Against this backdrop, a gradual integration of supervision into the clinical practice may pave the way to ensuring the effective delivery of targeted services. Furthermore, the inclusion of ideas or voices representing people for whom the service is designed can be useful. For example, a supervision process designed for the Rohingya people may include an advisory committee in order to represent the Rohingya service users voices. Suggestions or recommendations can be incorporated into the process to better fit the services in terms of the cultural sensitivity and appropriateness of the services.

> *Representation of people from the same community may ensure the suitability and accuracy of the services being offered.* (Senior Psychologist)

The exchange of cross-cultural knowledge about working with displaced communities or refugees can be a valuable addition to the supervision process. Contrary to professionals in low-resource settings, professionals in high-resource settings have access to various opportunities. These professionals can create a platform to exchange their unique experiences with professionals in other contexts. This exchange can create an opportunity to disseminate cross-cultural insights and lead to a shared understanding of what can be done to improve quality of care.

> *Regular meetings (experts and practitioners) can be arranged to exchange understanding, knowledge, and experience which may be helpful. Beneficiaries can be benefitted by this as well.* (Senior Psychologist)

As professionals in low-resource settings lack opportunities to improve their capacity for providing care, resources produced (e.g., translating articles into Bangla, the inclusion of local experts, and producing culture-specific manuals or therapies) in local languages can be beneficial, especially for those experiencing language barriers. Moreover, producing resources in a local language can sustain motivation. Cultural adaptations of evidence-based therapies can ensure a service takes into account the cultural aspects that determine the manifestations of and recovery from psychopathologies.

5.4.2. How to Make Supervision More Familiar for Syria

Usually in Arabic and Islamic cultures, people sit in a group and think together, e.g., in a mosque (Masjid), library, or school to learn the holy Quran, Arabic languages, and philosophy (Moral). They have a mentor (teacher/معلم) to learn from and with. A teacher is someone who has enough experience to guide and provide them with knowledge. This is reflective of Islamic culture and practice. A lot of famous imams have written books about how to know yourself, become self-aware of your feelings, face challenges, and manage everything (Ibad 2016). Religious leaders, especially in Islam, have conducted and used

reflective discussion. In the last century, a new perspective has dominated, focusing less on thinking, dialogue, or asking questions and more on keeping information in one's mind.

The Syrian practitioners who were involved in this paper consider local practices in addition to the suggested supervision model as community-based resources; this blend might be useful in bridging the gap between local and Western MHPSS practices.

Practitioners in the pilot supervision program believe that when designing any new clinical supervision program, it is important to set a clear goal for it, include educational and role-playing therapeutic techniques and methodologies, include staff care sessions, and use standardized terminology within the supervisory work.

> *Supervision should include teaching, application and role play around therapeutic techniques and methodologies. The sessions should include the aspect of taking care of themselves and pressure.* (Psychiatrist)

## 6. Discussion

In this paper, we critiqued the concept of WEIRDNESS embedded in MHPSSs in humanitarian contexts. We relied on our findings from the pilot supervision program and our reflective discussions to uncover the idiosyncratic contextual factors determining the supervision process. The knowledge we collectively produced highlighted the importance of local knowledge and familiarity in ensuring the relevance of supervision in a given context. As local MHPSS practitioners play a vital role in working toward collective community healing, we centred our discussions around depicting to what extent Western-based values and assumptions are relevant and helpful in two protracted displacement settings. WEIRDNESS is so entrenched in psychology that it was carried out in the area of supervision as well. As evident in our discussions, this approach did not benefit local MHPSS practitioners providing support to the displaced communities. On the contrary, it resulted in ambivalent feelings and thoughts among local practitioners. Therefore, to move away from WEIRDNESS, we came up with a decolonial supervision model called "WONDERFUL": Western-originated, needs-based, decolonial, exchange of knowledge and skills, respectful and reflective, flexible, useful, linking, and collaboration. Each component of the WONDERFUL approach is explained below (Table 1).

**Table 1.** The proposed model of supervision: WONDERFUL.

| Components | Description |
| --- | --- |
| Western-originated | Western-originated programs integrating and embracing Eastern ideas, knowledge, and perspectives. |
| Opportunity | An opportunity to defuse cross-cultural resources and findings. |
| Need-based intervention | Local knowledge, voices, understanding, and existing healing processes are acknowledged and taken into account. |
| Decolonial approach | Deconstruction of colonial ways of understanding research while embracing indigenous people's understanding and perspectives. |
| Exchange of knowledge and skills | Exchange of knowledge and skills produced in Western and Eastern contexts. |
| Respectful and reflexive | Ensure respect through increased collaboration and participation. |
| Flexible | Flexibility provides an opportunity to contribute to any part of the program based on expertise and experience. |
| Useful | Knowledge produced in a particular context is deemed useful and sustainable. |
| Linking and collaboration | Increased collaboration strengthens professional relationships. |

### 6.1. Western Origin

In our program, we acknowledge that the program originated in the West, having received funding from a resourceful, rich country. We recognize that both Western and Eastern regions have their own unique contributions to the world and during world crises. The program shares Western resources and advancement through different facilities of

UNSW. The program has secured funds from a Western-based country to implement the research in a humanitarian context.

### 6.2. Opportunities

We see this program as an opportunity to develop a different supervision approach across contexts that integrates the needs and characteristics of each context involved. We see this as an opportunity to share resources from each context, enable cross-cultural learning, and conduct research in each context utilizing Western resources. These include sharing ideas, learning from each other, and translating resources into Bangla and Arabic. This also includes a fortnightly online journal club to discuss research findings from different perspectives and contexts. We also discuss and create a supervision approach based on the understanding that there are different therapy models and skills required in each context.

### 6.3. Needs-Based Intervention

Humanitarian organisations often apply interventions based on colonized assumptions, e.g., that depression, anxiety, and PTSD are the dominant presentations of mental health problems in all contexts. In reality, the issues of each community are different. For example, PTSD is not considered common in some local contexts. Instead, our program avoided Western assumptions and focused on participatory and needs-based approaches in line with research that supports a needs-based approach (Chien et al. 2006). Our strategies included organizing stakeholders' workshops, as well as the involvement of practitioners, managers, service users/beneficiaries, local experts, and community members, to explore their perspectives to design the intervention. We also attempted to recruit local community volunteers as cross-cultural mediators and translators. We organized cross-cultural and local context training sessions for all the supervisors. We developed a handbook for the supervisors about MHPSS needs and supervision in each context.

### 6.4. Decolonial Approach

Humanitarian interventions have been criticised for their perpetuation of colonisation (Summerfield 2008). In order to decolonize mental health in humanitarian settings, the widespread practice and assimilation of decolonial approaches are necessary, including clinical supervision. The deconstruction of the WEIRD supervision programme involved strategies such as the inclusion of feedback from a Rohingya Advisory Committee (RAC), incorporating their responses as well as those from resulting from genuinely collaborative discussions with Syrian practitioners and beneficiaries. Our main focus was to incorporate local and indigenous ways of healing, the endemic manifestations of psychopathologies, and community perspectives, as well as to mobilize community resources into the new supervision program. The primary focus was for displaced people's voices to not go unheard and unheeded. Decolonization can bring about the repatriation of indigenous life (Tuck and Yang 2012). Our strategies included involvement of advisory groups for obtaining their ongoing advice, placing them rightfully in the expert position of their own experiences. We also ensured the participation and inclusion of community members and beneficiaries in both contexts as cross-cultural mediators for cultural and linguistic translation during training and supervision sessions.

### 6.5. Exchange of Knowledge and Skills

Our program focused on interchange and dissemination of ideas, experiences, stories, technologies, skills, and expertise between individuals or group of employees. Often Westerners from humanitarian organisations assume that they will transfer knowledge or educate those living in LMICs; however, in reality these societies over hundreds and thousands of years have been able to live with and overcome difficulties; they already have mechanisms and methods that help them in their own way. In our program, Western supervisors consider, reflect on, and learn from these practices so as to use them with the refugees they work with in their countries of asylum. On the other hand, local practitioners

benefit from Western, evidence-based approaches. We selected co-supervisors from both local and Australian contexts, allowing for an organic and reflexive space for all to grow.

### 6.6. Respectful and Reflexive

There was and continues to be a deep respect for the unique and diverse set of knowledge and skills that each member brings from each context. As team members, we consider ourselves to be connecting parts of a larger system and we are practicing being reflective on our actions, emotions, feelings, and frame of reference, as well as their potential impact on others. We make decisions through collaborative agreements about action points. Moreover, the supervisors are equipped with a guidebook, and they are advised to use their own reflections and observations according to the practitioners' needs during the session.

### 6.7. Flexible

We practice and promote a flexible approach to working rather than a structured format for both us and for our supervisors. We believe that this flexibility is suitable for the changing needs of practitioners and offers a way to develop the preferred model of supervision from a context. We assume that structured and formatted interventions would limit openness toward participants' needs and hinder the organic process of developing something new that would be more useful for and responsive to different contexts. Our strategies include recruiting and arranging international and local supervisors for the supervision sessions, which requires flexibility, adaptation, the accommodation of a varied range of expertise, training, and a culturally sensitive approach.

### 6.8. Useful

By useful, we mean that interventions should be useful for those who are working with a specific community in a specific context. In our case, these contexts are Syria and Bangladesh. Our intention was to design an intervention that is well-suited, customized, and useful for each context rather than design a fixed format and replicate it to different contexts. A co-supervisor design means that we can draw on the resources and knowledge of both supervisors: a local supervisor ensures appropriate adaption relevant to the context and an Australian supervisor supports with other resources, supervision training, and modelling.

### 6.9. Linking and Collaboration

We value collaborations and links with other organizations and professional bodies to deconstruct the program from a wired perspective. We also link and use resources from both Western experts and local specialists to provide high-quality interventions. We collaborate to provide the integrated model of supervision with guiding principles. We have an informal collaboration with the Suicide Prevention Working Group in the context of Cox's Bazar.

This participatory research project involves an ongoing, iterative process in which research informs practice and practice informs research (Abma et al. 2017). Rather than focusing on the objectivity of independent researchers, the participatory model considers the involvement of researchers in the implementation of programs as a strength rather than a weakness. Firstly, previous participatory research has shown that it is often not possible to fully separate researchers, participants, and practice when engaging in work which seeks to improve outcomes for a marginalized community (Goodkind et al. 2017). Second, engagement in qualitative research of this kind involves reflexive practice about the positionality of the researchers (Suffla et al. 2015). The reflective discussions provided space for the group to consider possible biases in their interpretations, to offer alternative points of view, and to explicitly address how learning from praxis can be integrated into theoretical models. For instance, through the reflexive discussion, we realized that the exchange of knowledge and resources is associated with an increased collaboration between Western

and Eastern professionals. Moreover, the discussions have increased the incorporation of indigenous practices, enabling the deconstruction of colonial approaches to viewing mental health issues and clinical supervision.

## 7. Conclusions

The aim of the paper was to provide an alternative model (i.e., WONDERFUL) of supervision that originates in Eastern contexts. This model enabled us to deconstruct WEIRD concepts of supervision while bringing back indigenous ideas and healing components. It is expected that studies using the proposed model of supervision will bring changes to humanitarian contexts, especially in LMICs, increasing clinical competence, reducing mental health problems, and improving service users' satisfaction.

**Author Contributions:** S.A.L. contributed to idea generation, development of the new model of supervision, and initial draft of the manuscript. M.O.F. contributed to idea generation, development of the new model of supervision, initial and final draft of the manuscript and did the revisions. S.J. contributed to idea generation, proposed the new model of supervision, initial draft of the manuscript. A.B. contributed to idea generation and initial draft of the manuscript. G.K. contributed to the initial and final draft of the manuscript and did the revisions. R.W. contributed to idea generation, development of the new model of supervision, initial and final draft of the manuscript and did the revisions. S.W. supervised and led the reflective discussion to generate ideas and develop the new model of supervision and contributed initial draft of the manuscript. All authors have read and agreed to the published version of the manuscript.

**Funding:** This research is funded by ELRHA's Research for Health in Humanitarian Crises (R2HC) Program (Grant Number: RG203720), which aims to improve health outcomes by strengthening the evidence base for public health interventions in humanitarian crisis. R2HC is funded by the UK foreign, Commonwealth and Development Office (FCDO), Wellcome, and the Department of Health and Social Care (DHSC) through the National Institute for Health Research (NIHR). The funding body had no role in the conceptualization; writing of the report; or the decision to submit the report for publication.

**Institutional Review Board Statement:** Ethical approval for the focus group discussions was provided by The University of New South Wales (UNSW) (HC220236) and University of Dhaka (IR211201).

**Informed Consent Statement:** We obtained informed consent from each participant involved in the study. Written informed consent has been obtained from the participants to publish this paper.

**Data Availability Statement:** FGD data will be available upon request from the corresponding author.

**Conflicts of Interest:** The authors declare no conflict of interest.

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
