# Peer review of "Clinical Supervision across Australia, Türkiye, Syria, and Bangladesh: From WEIRD to WONDERFUL"

_socsci, doi:10.3390/socsci12030170_

Round 1

Reviewer 1 Report

Dear Authors,

Thank you for the promising paper! By analyzing qualitative materials collected from focus group discussions and reflective discussions, the authors present a new model for clinical supervision. To improve the paper, I would like to ask you to pay more attention to the ways in which the methods and materials were used in the process:

Lines 11-16 are not describing your methods: focus group discussions and reflective discussions. This section should be revised completely (e.g., the number of participants was 19, not 32)

Lines 128-135. You should make it explicit that the first form of knowledge production is related to the focus group discussions. The second form has to do with the reflective discussions. Overall, the methodological approach could be seen as participatory research.

Lines 156-162. Since you have transcribed and translated the data acquired from the focus group discussions, you could present details (e.g., how many hours were recorded, how many pages were transcribed) and examples from the actual data (see below).

Lines 163-194. You are not explaining how the reflective discussions were analyzed (e.g., whether they were audio recorded, transcribed and analyzed).

Line 195. The manuscript has Introduction, Methods and Discussion. Would it make sense to call this section “Results”?

Pages 5-8: Add some examples to demonstrate your analysis.

I find it interesting that the academic home of WEIRD is the field of Psychology; it seems reasonable to say that Social Sciences cannot really avoid being labeled similarly (i.e., being Western, Educated, Industrialized, Rich and Democratic). Perhaps you should remind the readers of Social Sciences of WEIRDNESS in Social Sciences. 

Minor observations

Line 33: Sijbrandij et al., 2017 is missing from References

Line 50: Kemp et al., 2009 is missing from References

Line 89: Explain the abbreviation/modify the list of references: BBS

Line 100: UNHCR 2022 is missing from References

Line 126: ”WIEIRD”

Line 140. The mean age was 44.33/32.26? Consider rounding down.

Line 188. Delete

Line 268. First Bangladesh, then Syria.

Lines 281-282: ”practitioners working in Cox’s Bazar were fluent in English creating a significant barrier for effective communication.” Is there something missing?

Line 414: Explain the abbreviation: UNSW 

Line 429: The title should be bolded.

Lines 10, 82, 190, 262, 446, 449-551, 464, 467: typos / lack of spaces / lack of words / missing dot.

Line 435: Chien et al., 2006 is missing from References

Line 446: Explain the abbreviation: LMIC 

Author Response

Dear Authors,

Thank you for the promising paper! By analyzing qualitative materials collected from focus group discussions and reflective discussions, the authors present a new model for clinical supervision. To improve the paper, I would like to ask you to pay more attention to the ways in which the methods and materials were used in the process:

Lines 11-16 are not describing your methods: focus group discussions and reflective discussions. This section should be revised completely (e.g., the number of participants was 19, not 32)

Reply: We thank the reviewer for this comment. We corrected the number in the abstract:

Focus group discussions and reflective discussions included 19 participants involving both practitioners and authors across sites.

Lines 128-135. You should make it explicit that the first form of knowledge production is related to the focus group discussions. The second form has to do with the reflective discussions. Overall, the methodological approach could be seen as participatory research.

Reply: We thank the reviewer for this comment. We addressed the inquiry at Page 7, line between 1-10, highlighted in yellow in the manuscript. Here is the corresponding text:

We followed a participatory research approach that focuses on the process of sequential or successive action and reflection involving local people. This approach draws on local knowledge and viewpoints that are essential to form the basis of research and planning (Cornwall & Jewkes, 1995). In the present study, we engaged in two forms of knowledge production: focus group discussion and reflective discussion. Firstly, we conducted a series of stakeholder workshops with practitioners involved in the pilot as part of the C4C program. Second, we engaged in a series of regular structured meetings which involved reflective conversations aiming to give equal weight to all voices in the conversation. We used these conversations to synthesise information from the stakeholder workshops with our professional, cultural, and lived experience to generate a new model for supervision in humanitarian settings

Lines 156-162. Since you have transcribed and translated the data acquired from the focus group discussions, you could present details (e.g., how many hours were recorded, how many pages were transcribed) and examples from the actual data (see below).

Reply: We thanked the reviewer for this comment. We added the following details as follows below.

Focus group discussions (two in Northwest Syria/Türkiye and one in Bangladesh) were audio recorded and transcribed into Arabic and Bangla. Approximately six hours were recorded. The transcriptions were translated into English to work collaboratively in analyzing discussion data. The final transcripts were 10 pages for Syrian practitioners and 12 pages for Bangladeshi supervisors.

We also included relevant data extracts for our results. These were given as quotes of the participants and highlighted in the manuscript.

Lines 163-194. You are not explaining how the reflective discussions were analyzed (e.g., whether they were audio recorded, transcribed and analyzed).

Reply: We thanked the reviewer for this comment. We clarified this as follows and added it to the main text highlighted in yellow.

The reflective discussions involved collaborative analysis, drawing on the clinical experience of the group members, which resulted in the generation of the model.

Line 195. The manuscript has Introduction, Methods and Discussion. Would it make sense to call this section “Results”?

Reply: We thanked the reviewer for this comment. Following on this suggestion, we added the title of “Results” right before Line 195.

Pages 5-8: Add some examples to demonstrate your analysis.

Reply: We thanked the reviewer for this comment. We added the following part:

It is argued that basic cognitive and motivational processes vary across cultures, with evidence suggesting that people of different ethnic backgrounds tend to explain and predict behavior differently (Henrich et al., 2010a). Therefore, findings generated from a respective setting may not represent people from various culture, putting a question remark on the universality of the generalization (Henrich et al., 2010b). Against this backdrop, growing attention has been paid on deconstructing and localizing research by embracing indigenous perspectives and ideas.

I find it interesting that the academic home of WEIRD is the field of Psychology; it seems reasonable to say that Social Sciences cannot really avoid being labeled similarly (i.e., being Western, Educated, Industrialized, Rich and Democratic). Perhaps you should remind the readers of Social Sciences of WEIRDNESS in Social Sciences. 

Reply: We thank the reviewer for this comment. We edited the sentence and added a reminder for this point in other social sciences too. We did not want to extend the focus of this discussion to other social science disciplines but included a reminder considering the readers of the journal.

Psychology, as other social sciences, has been defined on research on WEIRD populations, Western, Educated, Industrialised, Rich and Democratic (Roberts et al., 2020).

Minor observations

Line 33: Sijbrandij et al., 2017 is missing from References

Reply: We added this to our references.

Line 50: Kemp et al., 2009 is missing from References

Reply: We added this to our references.

Line 89: Explain the abbreviation/modify the list of references: BBS

Reply: We explained it in the refence as “Bangladesh Bureau of Statistics”.

Line 100: UNHCR 2022 is missing from References

Reply: We added this to our references.

Line 126: ”WIEIRD”

Reply: We corrected this as “WEIRD”.

Line 140. The mean age was 44.33/32.26? Consider rounding down.

Reply: We modified this as follows:

The average age of practitioners was 44 in Northwest Syria/Türkiye and 32 in Bangladesh.

Line 188. Delete

Reply: We deleted it.

Line 268. First Bangladesh, then Syria.

Reply: We did as suggested.

Lines 281-282: ”practitioners working in Cox’s Bazar were fluent in English creating a significant barrier for effective communication.” Is there something missing?

Reply: We edited this sentence for further clarity.  

While some of the local practitioners had good or moderate command in English, the majority of practitioners experienced speaking English as a significant barrier for effective communication.

Line 414: Explain the abbreviation: UNSW. 

Reply: We gave the full name of UNSW here and used the acronym in the rest of the manuscript.

Ethical approval for the focus group discussions was provided by The University of New South Wales (UNSW) (HC220236) and University of Dhaka (IR211201).

Line 429: The title should be bolded.

Reply: We bolded the title.

Lines 10, 82, 190, 262, 446, 449-551, 464, 467: typos / lack of spaces / lack of words / missing dot.

Reply: We are sorry for these mistakes. We edited the text accordingly.

Line 435: Chien et al., 2006 is missing from References.

Reply: We added this to our references.

Line 446: Explain the abbreviation: LMIC. 

Reply: We explained this abbreviation as low- and middle-income countries here and use the abbreviation in the rest of the manuscript.

However, supervision received scant attention in humanitarian contexts in low-and-middle-income countries (LMIC).

Reviewer 2 Report

Page 1, line 44. it is a bit of an exaggeration to say that we can "heal a community."

It needs to be clarified. Did some of the authors of the study also participate in the intervention? Isn't this a research bias? The authors should consider this aspect or clarify the situation.

Page 2. "UN Human Rights Chief Points to 'Textbook Example of Ethnic Cleansing' in 77 Myanmar | UN News, n.d." it does not seem to be a correct citation in APA 7 format

There needs to be more than the theoretical part. The authors should explain why the professionals from Bangladesh, Syria, and Türkiye do not have sufficient professional supervision. It is clear that in these cultures, it is a big challenge from the point of view of mental health. However, it is unclear why, from the authors' perspective, the professionals are not trained well enough and would need this intervention. It must also be explained why this intervention proposed in the study could increase their work.

The authors must present in more detail what this intervention consisted of because it still needs to be clarified. I recommend that there be a protocol or guide that briefly presents the work sessions and the objectives.

In the scientific work, it is unnecessary to mention the psychologists (Scarlett Wong & Omar Faruk) who coordinated and what training they have in detail. You can mention anonymously that the two practitioners have training in clinical psychology and as a clinical supervisor or humanitarian contexts, without mentioning by name, because from a scientific point of view, it has no value. The authors can give them credit in the Acknowledgments section.

The study's objective is interesting and useful from a practical point of view, but it seems like it needs to be completed at the moment. The perspective I want to bring is that the study should be sent to the journal after investigating the real impact of the intervention proposed in the paper.

Author Response

Page 1, line 44. it is a bit of an exaggeration to say that we can "heal a community."

Reply: We edited the sentence as follows:

Supervision provides an opportunity to support these staff to alleviate the psychosocial suffering of members of their community.

It needs to be clarified. Did some of the authors of the study also participate in the intervention? Isn't this a research bias? The authors should consider this aspect or clarify the situation.

Reply: We acknowledged the potential research bias and added the following explanation to clarify our position.

This participatory research project involves an ongoing, iterative process in which research informs practice and practice informs research (Abma et al., 2017). Rather than focusing on the objectivity of independent researchers, the participatory model considers the involvement of researchers in the implementation of programs as a strength rather than a weakness. Firstly, previous participatory research has shown that it is often not possible to fully separate researchers, participants, and practice when engaging in work that seeks to transform outcomes for a marginalized community (Goodkind et al., 2017).  Second, engagement in qualitative research of this kind involves reflexive practice about the positionality of the researchers (Suffla et al., 2015). The reflective discussions provided space for the group to consider possible biases in interpretation, offer alternative points of view, and explicitly address how learning from praxis can be integrated into theoretical models. For instance, through the reflexive discussion, we realized that the exchange of knowledge and resources is associated with increased collaboration between Western and Eastern professionals. Besides, it has widened the horizon of incorporating indigenous practices enabling the deconstruction of colonial approaches to view mental health issues and clinical supervision. 

Page 2. "UN Human Rights Chief Points to 'Textbook Example of Ethnic Cleansing' in 77 Myanmar | UN News, n.d." it does not seem to be a correct citation in APA 7 format

Reply: We thank the reviewer for bringing this to our attention. We have corrected the reference.

There needs to be more than the theoretical part. The authors should explain why the professionals from Bangladesh, Syria, and Türkiye do not have sufficient professional supervision. It is clear that in these cultures, it is a big challenge from the point of view of mental health. However, it is unclear why, from the authors' perspective, the professionals are not trained well enough and would need this intervention. It must also be explained why this intervention proposed in the study could increase their work.

Reply: We added the following sentences under the descriptions of the context.

Besides, the majority of the organizations providing MHPSS service in humanitarian contexts do not have provisions for clinical supervision partly due to the lack of awareness of its potential benefits. Besides, the lack of mental health professionals with experience in clinical supervision has also contributed to the gap.

However, similar to Bangladesh's context, supervision is often overlooked because of the restrictions on funding and lack of awareness about its potential benefits.

The authors must present in more detail what this intervention consisted of because it still needs to be clarified. I recommend that there be a protocol or guide that briefly presents the work sessions and the objectives.

Reply: We thanked the reviewer for this comment. We explained our C4C program in detail in the manuscript. We provided the details related to the aim and content of the program, delivery method, and session characteristics. Please see the description added to the manuscript below and highlighted in yellow at Page 3 and 4.

Learnings from this program were used to develop a subsequent scaled-up program – the Caring for Carers (C4C) Program. Participatory research and equal representation are the crucial ethos of the C4C program. The program aims to test the effectiveness and acceptability of an online clinical supervision program in the humanitarian contexts of Bangladesh, Northwestern Syria, and Türkiye to improve the well-being of practitioners, increase their efficacy in clinical skills, and improve service quality and satisfaction. The C4C program consists of two terms of 8 months long supervision sessions provided via an online platform on a fortnightly basis to a group of MHPSS practitioners. Each session lasts 90 minutes. One Australian and one local supervisor (Bangladeshi or Syrian) co-facilitate the supervision group of four to six MHPSS practitioners. The program focuses on reflective practice, case discussion, and skills learning and improvement while providing MHPSS services to the displaced communities in the respective context.

In the scientific work, it is unnecessary to mention the psychologists (Scarlett Wong & Omar Faruk) who coordinated and what training they have in detail. You can mention anonymously that the two practitioners have training in clinical psychology and as a clinical supervisor or humanitarian contexts, without mentioning by name, because from a scientific point of view, it has no value. The authors can give them credit in the Acknowledgments section.

Reply: We modified the related section in light of this suggestion. We removed the names from this section.

The study's objective is interesting and useful from a practical point of view, but it seems like it needs to be completed at the moment. The perspective I want to bring is that the study should be sent to the journal after investigating the real impact of the intervention proposed in the paper

Reply: We thanked the reviewer for the constructive feedback and suggestions. We believe that this paper outlines the process of our suggestions to transform clinical supervision from Western to locally focused. Although this paper is very embedded in our wider supervision program, C4C, it is different from the rest of our program, which mainly focuses on testing the effectiveness of online clinical supervision programs based on local knowledge and values. This paper can be considered the first step to form the base of our supervision program, and its effectiveness will be tested later.

Reviewer 3 Report

Thank you for the opportunity to review this very interesting paper that offers valuable insights into supervision in an important contextual reality. This work adds to an area in need of research and clinical development.

There are some changes that I think are needed to enhance the paper and its readership.

1. The C4C program referred to on p.2 needs some better description so that readers unfamiliar with it can understand its value and why referenced here.

2. WEIRD Psychology - this is a very good point. I tend to favour citing original authors when presenting these contextual theoretical understandings. Thus, I suggest you consider referencing the work of Henrich, Heine & Norenzayan 2010 - one paper is in Nature and the other in Behavioural and Brain Sciences.

3. At line 381, The prophet Mohammed - the sentence is awkward - I suggest rewriting 

4. The paragraph starting at line 463 has a long run on sentence - please revise.

5. The section on Wester Origin is interesting but the bullet form of presentation seems less engaging. It might be presented in chart format with a discussion for each area (as an example). 

6. The paper lacks a conclusion / discussion 

7. The paper lacks an  ethics statement.

Author Response

Thank you for the opportunity to review this very interesting paper that offers valuable insights into supervision in an important contextual reality. This work adds to an area in need of research and clinical development.

There are some changes that I think are needed to enhance the paper and its readership.

The C4C program referred to on p.2 needs some better description so that readers unfamiliar with it can understand its value and why referenced here.

Reply: We thanked the reviewer for this comment. We added the details of our program at Page 3 and Page. The description is:

Learnings from this program were used to develop a subsequent scaled-up program – the Caring for Carers (C4C) Program. Participatory research and equal representation are crucial ethos of the C4C program. The program aims to test the effectiveness and acceptability of an online clinical supervision program in the humanitarian contexts of Bangladesh, Northwestern Syria and Türkiye to improve well-being of practitioners, increase their efficacy in clinical skills, improve service quality and satisfaction. The C4C program consists of two terms of 8 months long supervision sessions provided via an online platform on a fortnightly basis to a group of MHPSS practitioners. Each session lasts 90 minutes. One Australian and one local supervisor (Bangladeshi or Syrian) co-facilitates the supervision group of four to six MHPSS practitioners. The program focuses on reflective practice, case discussion, and skills learning and improvement while providing MHPSS services to the displaced communities in the respective context.

WEIRD Psychology - this is a very good point. I tend to favour citing original authors when presenting these contextual theoretical understandings. Thus, I suggest you consider referencing the work of Henrich, Heine & Norenzayan 2010 - one paper is in Nature and the other in Behavioural and Brain Sciences.

Reply: We thanked the reviewer for this suggestion. We cited the suggested papers. Please see the edited text below.

Majority of psychological research is dominated by the inclusion of white subjects, without attending to the impact of racial and cultural diversity (Roberts et al., 2020; Henrich et al., 2010a).

Therefore, findings generated from a respective setting may not represent people from various culture, putting a question remark on the universality of the generalisation (Henrich et al., 2010b).

At line 381, The prophet Mohammed - the sentence is awkward - I suggest rewriting.

Reply: We thanked the reviewer for this comment. We rewrote this part for further clarification.

Religious leaders, especially in Islam have done and used this reflective discussion.

The paragraph starting at line 463 has a long run on sentence - please revise.

Reply: We revised it and highlighted the revised sentence in yellow.

The section on Wester Origin is interesting but the bullet form of presentation seems less engaging. It might be presented in chart format with a discussion for each area (as an example). 

Reply: We thanked the reviewer for this suggestion. We added a table delineated each component clearly.

The paper lacks a conclusion / discussion 

Reply: We added a conclusion section at the end of the manuscript.

Conclusion

The aim of the paper was to provide an alternative model (i.e., WONDERFUL) of supervision originated in eastern contexts. This model enables us to deconstruct WEIRD concepts of supervision while bringing back the indigenous ideas and healing components. It is expected that the study with the proposed model of supervision will bring change in humanitarian contexts, especially in LMICs in pursuit of increasing clinical competence, reducing mental health consequences, and improving beneficiary satisfaction.

The paper lacks an ethics statement.

Reply: We added the ethics statement.

Ethics Statement

Ethical approval for the focus group discussions was provided by The University of New South Wales (UNSW) (HC220236) and University of Dhaka  (IR211201).

Round 2

Reviewer 3 Report

Thank you for the attention to the revisions.